# Recapturing the Oral Tradition of Storytelling in Spiritual Conversations with Older Adults: An Afro-Indigenous Approach

Florence Akumu Juma 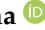

Martin Luther University College, Wilfrid Laurier University, Waterloo, ON N2L 3C5, Canada; fjuma@luther.wlu.ca

**Abstract:** The historical value of the oral tradition permeates literature as represented in multiple disciplines, including theology. An aspect of this tradition has proven viable in spiritual conversations with older adults. This paper will discuss the oral tradition's medium of storytelling and listening to demonstrate its relevance in therapeutic conversations with older adults. Therapeutic storytelling is a distinct intervention prevalent in the African oral tradition This approach is also gaining attention in the contemporary context, blending seamlessly within the narrative approach. Using the quantitative research method of ethnography and autoethnography, I analyze specific therapeutic encounters that employ a storytelling/listening approach in spiritual conversations. The analysis reveals the relevance of storytelling in specific therapeutic encounters. Storytelling is gaining interest and reclaiming space in therapeutic settings with diverse populations, but mostly with older adults. The study also highlights the apparent similarities between the traditional approach to storytelling and the narrative approach in the contemporary therapeutic milieu.

**Keywords:** African oral traditions; oral traditions; storytelling; narrative approach; spiritual care; older adults; geriatric care

## 1. Introduction

The oral tradition is a global phenomenon. The human species have relied on the multidimensional aspect of the oral tradition for centuries for the transmission of culture and history, among other disciplines. Historian John Foley states, "[the] oral tradition has been an ancient human tradition found in 'all corners of the world'" (Mackay 1999). Although the emergence of literary tradition made a significant contribution to the explosion of information (cf. Toffler 1970), it did not fully replace the oral tradition. The two traditions have coexisted, assuming complementary roles in the process of transmitting knowledge and connecting individuals and communities across generations and geographic regions. As Foley posits:

> Indeed, if these final decades of the millennium have taught us anything, it must be that oral tradition never was the other we accused it of being; it never was the primitive, preliminary techno logy of communication we thought it to be. Rather, if the whole truth is told, oral tradition stands out as the single most dominant communicative technology of our species as both a historical fact and, in many areas still, a contemporary reality.

A simple definition of oral tradition is "The spoken relation and preservation, from one generation to the next, of a people's cultural history and ancestry, often by means of storytelling" (Oral Tradition n.d.). It is the "stories, beliefs, etc., that a group of people share by telling stories and talking to each other" (Merriam-Webster n.d.). In the pre-modern African context, the oral tradition found expression through multiple media, including proverbs, songs, stories, customary laws, and language (cf. Merrie 2019). Storytelling seems

to have emerged as one of the most organic and natural expressions, permeating all the other methods. Individuals engaged in storytelling/listening to connect, inform, admonish, educate even challenge each other as necessary. One unique aspect of storytelling was its therapeutic function peculiar within the African worldview. I have often borrowed from this therapeutic function of the African oral tradition as a bridge in the delivery of spiritual care in the contemporary health context. My interest grew from years of work with patients in a residential treatment facility for older adults with physical and cognitive decline. I recalled scattered incidents of observing the practice within my community of origin and participating in the tradition intermittently during school holidays or reading about it in books. I then purposed to explore further, thus my ongoing attempts at self-discovery and a desire to share the knowledge. This approach can add value to therapeutic conversations with all populations and most particularly with older adults. Used as an agent of healing, storytelling and listening can enhance communication in spiritual conversations among older adults in therapeutic contexts.

The African oral traditions pre-date the emergence of modern religious movements in Africa; yet, they seem to lend themselves well to various religious practices and expressions of worship, predominantly the Christian faith. As much as the African oral traditions were the method for transmission of culture, information, and knowledge from generation to generation, they also served to communicate and transmit spiritual practices and the tenets of faith. Such spiritual practices continue to the present time, conveying nourishment and healing for the soul (Juma and Juma 2017). Employing such a narrative language allows simplicity in [spiritual] dialogue through reflection on experiences rather than on abstract constructs (cf. Kenyon and Randall 1999). In the chapter "the church and the streets: an ethnographic study of the Christian hip hop scene in central Texas", Burnaby quotes Wilson (In Schneider and Kotarba 2016, p. 155) who writes, "most African societies place great worth in oral tradition because it is a primary means of conveying culture". It is also a mode of transmitting feelings and attitudes. Songs and music formed an integral part of the African oral tradition, which caused the two to be interrelated. Songs were employed to enhance certain aspects of the narrative and provide a rhythm for both the storyteller and the audience. Therapeutic storytelling is predominantly inherent in contexts where the sacred intertwines with the secular in all aspects of life. There are apparent similarities between this approach and the narrative approach in the contemporary therapeutic milieu. In this respect, a narrative approach is particularly suited to the exploration of such topics as meaning, spirituality, and wisdom and the connections they share (Bohlmeijer et al. 2011; Kenyon and Randall 1999). Storytelling was a cherished method of sharing knowledge, socializing communities, and promoting health and wellness in most of Sub-Saharan Africa (Finnegan 2003). It would seem that storytelling, similar to some of the transgenerational and multigenerational approaches in family therapy, can be recaptured in therapeutic conversations with all generations, but mostly with the older populations.

## 2. The Value of Storytelling in Therapeutic Contexts with Older Adults

The oral tradition's therapeutic approach to storytelling can serve as a viable strategy in spiritual conversations with older adults to facilitate meaning making (cf. Mitty 2010). Unless care practitioners develop ways to incorporate storytelling in their work with older adults, such stories remain buried in the memories of the clients, patients, and residents as they approach the final transition in life. Writing from the practitioner's perspective, Bergner (2007) outlines some beneficial features of using stories in psychotherapy, concluding his discussion with the statement, "stories are powerful devices for assisting patients" (p. 16). Practitioners can begin to shed light on those stories in their efforts to facilitate making meaning and foster healing in therapeutic settings. Although the broader oral traditions served humanity in the transmission of history, stories, folktales, and religious beliefs from generation to generation; therapeutic storytelling was a distinct intervention prevalent in Indigenous approaches to healing within diverse African communities (cf. Juma 2015; Akunna 2015). However, with increased immigration and the process of regulating the

helping profession in the contemporary context, some of the pre-modern approaches in therapy are likely to fade. Thus, the need to recapture relevant aspects of these traditions and weave them into the postmodern family therapy approaches.

Kerr et al. (2014) wrote about the value of storytelling in the health context, illustrating her point in an article on "Families, healing systems and journal" She reflected on her experience stating:

> When I take my son to the doctor, we untangle the web of distortions provided by his birth parent in his younger years. We strive to make right the wrongs, to set his story straight. I'm grateful his new doctor knows the patient's story matters—and that it matters who's telling the story, who's listening.

Moen (2006) also concludes his discussion by stating, "Thus, storytelling as a way of recounting and creating order out of experience starts in childhood and continues through all stages of our lives" (p. 60).

*2.1. The Art of Storytelling*

Although the oral tradition finds expression through multiple media, storytelling stands out in this work as the dominant medium; as such, the need to look closely and formulate a definition of the term. Numerous definitions of the term storytelling exist, all of which share common elements, including communicating events using words, images, and sounds with some improvisations (cf. Haigh and Hardy 2011). The National Storytelling Network (National Storytelling Network (NSN) n.d.) provides an elaborate description of the word storytelling, including the nature, method, and goal. The author acknowledges that the word "storytelling is often used in many ways and give what is considered the understanding of the NSN in America. They present a basic understanding of storytelling "as an ancient art form and a valuable form of human expression" (https://storynet.org/what-is-storytelling/, Retrieved on 20 April 2022). Storytelling is a prevalent method of communication in various forms, including mainstream and social media. It is one of the most basic modes of communication and an integral part of being human. Through storytelling, people can find a deep connection as their stories weave into their separate lives, unveiling their common humanity. Storytelling can enable individuals' effortless and natural communication as people encounter each other. The pre-modern African communities relegated professional storytelling to older adults, who were known to be reservoirs of collective stories. Providentially, they were at a season in life that afforded them all the time needed to share those stories in various modes of communication. Storytelling seemed most efficient in transmitting knowledge, culture, and history to the young and emerging generation. In the contemporary context, practitioners need to be intentional and make time for the stories to evolve and serve their natural purpose-connecting people. Storytelling has the potential to serve as a medium that enables both narrator(s) and listener(s) to connect through shared stories.

As an art, storytelling can be acquired through structured education and training programs at various levels prescribed in relevant disciplines (cf. Meretoja 2017; Scott 2011; Cronon 2013). Individuals can learn and acquire storytelling skills and tools necessary to apply the method in research and practice. Writing from a business context, Taylor (2021) acknowledges the effect of storytelling on one's identity and challenges employers to encourage the culture of innovative storytelling among their employees. As a science, storytelling can be studied for clarity and consistency (cf. Taylor 2021; Cronon 2013; Houston et al. 2011; McBeth et al. 2005). In his presidential address, the president of the American Historical Association, William Cronon (2013), posed this question to his audience, asking, "What is the future of history?" He situated this question with a comprehensive introduction to provide the context stating:

> "In a distracted world where even undergraduates at top universities are increasingly challenged to read the kinds of books we have traditionally written, and at a moment when there seems to be widespread public doubt about whether to

continue supporting the study of the past as this organization has traditionally understood that activity, what is the future of history?"

Cronon then observes that there are many answers to the question and responds with what he considers the most basic answer, writing, "there is one answer that is arguably the most basic of all, and that is, simply storytelling". He elaborates his answer further by inviting his audience to reflect on the discipline. He posits:

> "We need to remember the roots of our discipline and be sure to keep telling stories that matter as much to our students and to the public as they do to us. Although the shape and form of our stories will surely change to meet the expectations of this digital age, the human need for storytelling is not likely ever to go away. It is far too basic to the way people make sense of their lives—and among the most important stories they tell are those that seek to understand the past". (p. 5)

Storytelling, therefore, is an art and science that is broad, encompassing multiple disciplines. The use of storytelling in research has also drawn criticism from qualitative researchers and can be considered work in process with promising results (cf. Houston et al. 2011). This work will adopt the understanding of storytelling from its most basic and oral traditional roots, the informal sharing of stories between two or more individuals through voice and gestures to connect and make meaning of their situations. In her book "Composing Life", the renowned anthropologist Mary Catherine Bateson (2001) wrote what seems to be a timeless quote. Bateson wrote:

> Wherever a story comes from, whether it is a familiar myth or a private memory, the retelling exemplifies the making of a connection from one pattern to another: a potential translation in which narrative becomes parable and the once upon a time comes to stand for some renascent truth. This approach applies to all the incidents of everyday life: the phrase in the newspaper, the endearing or infuriating game of a toddler, the misunderstanding at the office. Our species thinks in metaphors and learns through stories.

The therapeutic role of storytelling has been and can be therapeutic for both the narrator and the listener. This work focused on the therapeutic aspect of storytelling with older adults in a health care setting. The goal is to highlight the value of storytelling among older adults to facilitate spiritual conversations. This has been possible partly due to my unique professional context. I have worked with older adults in a specialized residential geriatric treatment facility, and the case analyzed below is representative of more than 10 years of work with this population.

### 2.2. Spiritual Conversations

What exactly constitutes a spiritual conversation? The answer to this question can be as varied as the understanding of the word 'spirituality'. In this work, the understanding of a spiritual conversation is equivalent to a therapeutic conversation conveyed in a language that takes into consideration relevant spiritual disciplines and seeks to foster an environment for Meaning-making within the framework of spiritual care. The word spiritual within the discipline and practice of spiritual care in contemporary use comes loaded with a historical understanding of the term 'pastoral care'. Although the classic term 'pastoral' may seem to convey an undertone or a focus of the caregiver or practitioner in the caregiving relationship, the more contemporary term 'spiritual', on the other hand, seems more descriptive of the care recipient or the layperson (client, congregant, patient, and resident) in the caregiving relationship. Indeed, the word spirit itself has multiple understandings and is frequently used in diverse contexts to convey different meanings for different purposes. The use in this work relates solely to its personal use—the essence of a human being that holds life. As defined in the Lexicon Oxford Dictionary 'spirit' is "the non-physical part of a person which is the seat of emotions and character" (https://www.lexico.com/definition/spirit, accessed on 8 April 2020). The dictionary

further describes the spirit as "[t]he non-physical part of a person regarded as their true self and as capable of surviving physical death or separation". Gleaning from the definition of the word 'spirit' above and the relationship that the dictionary made with the word religion, it makes sense that spiritual care has a long history in the care provided by religious leaders that sometimes the terms spiritual and religion tend to be used to refer to the same concept. A working definition of spiritual care is likely to encompass some understanding of religion in its broader sense beyond the confines of the institution of religion. This work is based on a working definition of spiritual care informed by the contributions of various researchers and authors in the field of spiritual care (cf. Ivy 2019; Boettcher 2018; Breitbart et al. 2018; Callahan 2017; Doehring 2015). The working definition of spiritual care for this work is—The care that seeks to sustain, support, and vitalize the health and wellbeing of the human spirit, aiming to promote holistic health and finding relevant treatment approaches to facilitate healing for a wounded spirit. Spiritual care integrates relevant therapeutic theories to instill hope and promote holistic health and wellbeing in individuals and their families from the inside out. This care is aimed at vitalizing the spirit to embolden the body and support both intra- and interpersonal relationships.

This definition informs the understanding of the word spiritual, 'the seat of emotions', 'true self', and 'surviving physical death'. Recognizing that spirit is also used to describe those qualities regarded as forming the definitive or typical elements in the character of a person, nation, or group, or in the thought and attitudes of a particular period. A spiritual conversation, therefore, is a conversation intended to engage in meaning-making, broad enough to involve one's beliefs and religious practices and transcending the physical. Such a conversation connects two or more individuals at a deeper emotional level. One of the approaches to achieving the goal of spiritual care is through spiritual conversations—talk therapy. Storytelling and listening lend themselves well to this approach of spiritual care and, more so, care for older adults (cf. MacKinlay 2006).

Historians have addressed the value of storytelling from various perspectives, including Mitty (2010, p. 58), who contends, "storytelling can be therapeutic. For the person," She proceeds, "it is both validating and valuing—as nothing else can do. There is a connection between old age and spirituality and a quest for transcendence—to express one's self as part of the human condition." Older adults who tell their stories keep their memories and values alive; it can reunite a family. Sometimes couched in terms of creativity, storytelling is sharing and transmitting something that is likely new, not known (or seen) before, and valued. Saffran (2021) writes Stories are not just ubiquitous; they are powerful. Imagery, verisimilitude, and the sense of a storyteller speaking in their own voice engage our emotions and stick in our memory. The question is, how do we go about employing storytelling and listening with older adults or encourage storytelling in therapeutic contexts? The answer to that question is the burden of this work. I identified a suitable methodology to analyze some of the work already performed in attempts to recapture this age-old tradition.

## 3. Methodology

This study will utilize data obtained through the qualitative method of ethnography and autoethnography from my work with older adults in various clinical programs. The data were readily available and organically obtained from therapeutic encounters with older adults and ongoing research on the therapeutic aspects of the African oral traditions. I will present a case study of an encounter with a patient in a specialized residential treatment facility for analysis and follow it with a discussion to highlight the value of storytelling in establishing therapeutic relationships, assessment, and treatment in spiritual care. Utilizing a qualitative research method of ethnography and autoethnography (cf. Nolan 2018), I analyze select spiritual care conversations that draw from this approach of the oral tradition as applied in my work with one older adult in a health care setting as a case example. In his study "Making our choices", George Fitchett (2011) makes an appeal for case studies stating, "Chaplains need these case studies to provide a foundation for further research about the efficacy of chaplains' spiritual care" (cf. Nolan 2018, p. 3).

### 3.1. Ethnography

Ethnography is defined as "the study and systematic recording of human cultures; . . . a descriptive work produced from such research" (Merriam-Webster n.d.). Dharamsi and Charles (2011) also describe ethnography as " . . . a research method that is used to gain a deeper understanding of human behaviour, motivation, and social interaction within specific and complex cultural contexts". They state, "ethnographies provide an in-depth reflection and analysis and paint a portrait of the ways in which culture-sharing groups interpret their experiences and create meaning from their interactions" (p. 378). This description denotes that the work of the ethnographer provides an in-depth view of the social, political, and cultural context within the cultural group. Ethnography is a qualitative research method where the researcher enters into the setting of those being researched to study and record culture as it unfolds. In his book "Qualitative Research", Shank (2002, p. 56) also accredits ethnography as one of the oldest and most vulnerable of all qualitative research methods. The term is derived from a Greek word that refers to a description of a group of people and their way of life (Suzuki et al. 2005).

### 3.2. Autoethnography

Autoethnography refers to the qualitative research where an author uses self-reflection and writing to explore their personal experience and connect this autobiographical story to wider social, cultural, and political meanings and understandings. I approach this study as a narrator of a cultural phenomenon with which I have basic knowledge and desire to reflect and understand further through research.

In their opening statement, Ellis et al. (2011) define autoethnography as "an approach to research and writing that seeks to describe and systematically analyze personal experience in order to understand cultural experience" (Abstract). Despite the challenges, and as Suzuki et al. argue, ethnography and autoethnography have merit in the field of qualitative research. They state that:

> "ethnographically informed methods can enhance counseling psychology research conducted with multicultural communities and provide better avenues toward a contextual understanding of diversity as it relates to professional inquiry" (p. 206)

In his work titled "Autoethnography in chaplain case study research", Nolan (2018) opens his discussion by making a case for case study research in spiritual care. He posits: "possibly the greatest value of case studies is their ability to take us into the intimacy of the bedside and allow fellow chaplains, healthcare colleagues and those who commission chaplaincy care, as well as the general public, to see what actually goes on in the private space of the chaplaincy/spiritual care relationship" (p. 11). Nolan acknowledges the challenges of methodology that meets the requirement for a "robust and logical" process that other can "check that what has been found out is credible and valid" (p. 12). He concludes his discussion by affirming the accessibility of research in a case study for both novice and seasoned researchers. I have used ethnographic and autoethnographic methods, incorporating a self-reflective approach, to analyze cumulative spiritual conversations that highlight the use of storytelling/listening with older adults in therapeutic settings. As stated above, the choice of this approach was determined, in part, by the availability of the data from ongoing work with older adults across diverse clinical programs and a curiosity to re-engage the oral traditions in spiritual care. I undertook this work with the understanding that in ethnography, researchers cannot claim to provide a "neutral account of others' experiences". As Dharamsi and Charles (2011) point out, the research is "influenced by a cultural mediated world, caught up in webs of significance [we] ourselves have spun" (p. 379). I also attempted a comparative analysis with the narrative therapy approach. Lee et al. (2014) concludes their study by stating that there are significant benefits for researchers and narrative research if narrative research allows for and affords story sharing.

My experience with storytelling accrued from learning about the practices and approaches of pre-modern communities in the continent of Africa. However, my ongoing interest and research are narrowed to specific approaches practiced by one indigenous group in Nyanza Province, within the shores of Lake Victoria in Kenya, namely, The Luo tribe. The Kenyan Luo tribe is a subgroup of the larger Luo community that spans the Eastern African countries of Uganda, Tanzania, Sudan, Congo, and Ethiopia (cf. Dietler and Herbich 1993). The tribe is the third largest community in Kenya and comprises 13 percent of the Kenyan population. Tradition has it that the Luo people traveled along the River Nile to settle in the Nyanza region on the shores of Lake Victoria in Kenya. As reflected through an interdisciplinary approach in contemporary health care contexts, traditional healing approaches in Africa are varied. Storytelling played a significant role in complementing other treatment methods. Such other methods were prevalent not just among the Luo but also among other communities as well as across the continent. Some examples include Cumes (2013), who introduced his paper on South African Indigenous healing stating that "Sangomas or inyangas are shamans, healers, priests, and prophets that have been the backbone of Bantu communities, especially in the rural south Africa for eons" (p. 58). Wallace (2003), who writes from a specific cultural group in Namibia, asserts that "Otjiherero-speakers [of Namibia] used herbal medicines, massage and specialist healers" (p. 356). Finally, Akunna (2015), in her article observing a practice of a tribe in Nigeria, asserts that "the Igbo people have revealed a predisposition to harnessing dance's creative force toward safeguarding the wellbeing, interests, and integrity, not only of individuals, but of the fabric of societal life" (pp. 39–40).

In this particular work, I present a case study of my ongoing work with older adults in a health care context for analysis. It represents my attempt at replicating what I have observed and experienced within specific groups in the African communities in diverse contexts and mostly among the Luo tribe as they revisit shared stories that helped community members make meaning of their experiences to facilitate the healing. It has been observed that some of these practices were meant to perpetuate a self-healing culture within the community (Juma and Juma 2017). The larger Kenyan Luo communities are spread over a wider region beyond the Northern Nyanza area—the location of my ancestry. Autoethnography as a method of choice in this study provided my Luo ancestry and my curiosity to learn more about the method. My grounding years consisted of limited exposure to the Luo culture with a mix of what was then emerging as a sub-culture with the traceable and dominant influence of the Christian faith. As such, I had the privilege of acquiring knowledge (sometimes as an observer) of this cultural group and sought to revisit what I knew vaguely to understand it more fully. I then accessed the knowledge, now rediscovered, to provide a personal and reflective (autoethnography) perspective (Shank 2002, p. 60). A unique feature of this method is spontaneity, requiring minimal to no structure in practice. The value includes expanding opportunities to establish rapport for deeper connection and gain pertinent information that informs assessment and ongoing interventions.

## 4. A Case Analysis

The following is an analysis of a typical scenario prevalent among the older adult in geriatric clinical programs in health systems and long-term care facilities. This case is representative of one among the many diverse populations likely to be encountered by spiritual care practitioners (SCPs) in those settings. I provide a brief description of the case and then capture a representative spiritual conversation with a resident through storytelling. A theory integration of the case follows with a comparative analysis of storytelling and narrative approaches.

### 4.1. Description

This case will follow an older adult living with a diagnosis of moderate to severe dementia and needing full support with daily tasks and personal care around the clock. Such support may be provided at home with the assistance of family and community

support staff or in a specialized long-term care facility. This individual may require admission to a health care facility occasionally for monitoring and treatment as necessary. The case will focus on early encounters following admission to a specialized treatment facility. The goal is to establish a therapeutic relationship, initiate assessment, and plan relevant treatment. The verbatim used is contrived based on actual spiritual conversations over a period of time with the patient. The SCP needs to be intentional in its approach and foster storytelling during all therapeutic encounters. For the purpose of this study, the patient will be named "Bob" C. The initial encounter occurs upon admission of Mr. C to a residential treatment facility. The SCP employs a storytelling approach to obtain information that can help in the process of learning to know Mr. C better, establishing a rapport to proceed with the assessment, and relevant interventions during the admission.

*4.2. Storytelling as a Medium in Establishing a Therapeutic Relationship*

The verbatim that follows will demonstrate the use of storytelling with an incoming patient. Mr. C was newly admitted to a residential treatment program for older adults living with a diagnosis of dementia. He arrived from home, brought in by his family, who has managed his care at home with the help of home and community support teams. His care has been manageable so far, with the assistance of community support. With time, his needs seemed to have exceeded the resources and supports put in place by community treatment teams, thus the recommendation for residential treatment in the geriatric program of a health care facility. Mr. C arrived with an elaborate medical history from his primary physician, complemented with notes from all the care partners who are providing his care in the community. On admission, the family accompanied Mr. C and met with the residential treatment team to provide additional information and complement the medical history as captured on paper. The spiritual care practitioner is part of the Admissions Assessment Meeting Team. This process is helpful in obtaining Mr. C's spiritual history and current spiritual/religious affiliation and practices.

Within two weeks of admission, the SCP plans 2–3 informal meetings as needed with Mr. C to obtain his story as narrated by Mr. C himself. This is where the informal nature of the oral tradition approach comes in handy. The SCP plans visits with Mr. C. at various times of the day to help identify the most suitable time. It is up to the SCP to facilitate a conducive environment and maximize the opportunity for storytelling. Below is a summary of typical conversations that help the SCP learn to know Mr. C and begin establishing a therapeutic relationship. I observed a practice among the identified community used for introductions at communal gatherings. It was a common practice in social, religious, or even political events. The lead person invites individuals with the question, "what is your story?" The invitation asks for much more than a simple who are you or what is your name? What is your story—calls an individual to stop and think about relevant aspects of their life that they would like to share as an introduction. The introductory conversation below is informed by that broader concept; however, it also acknowledges the need for cultural self-awareness (cf. Fitchett 2011). Thus recognizing the uniqueness of the patient/client and their different worldview. As such, I employ a modified approach with the same idea.

4.2.1. Storytelling 1: Learn to Know Mr. C

> SCP-1a: Hello
> P-1a: Hello
> SCP-2a: My name is Florence (pause)
> P-2a: Good
> SCP-3a: May I ask your name, Sir (pause)
> P-3a: My name is "Bob"
> SCP-4a: Bob that is a great name. Pleased to meet you Bob
> P-4a: And you too. What are you doing here?
> SCP-5a: I am here to listen
> P-5a: Listen to who?

SCP-6a: To you, if you let me. Is that okay with you?

Silence

P-6a: hum, I don't know. Look at me . . . I don't even know where I am or why am here. I'm just waiting for my family to come and take me back home.

SCP-7a: Home. (Pause). Home sounds like a good place. Where is home?

P-7a: My home . . . and the story begins

### 4.2.2. Storytelling Option 2: The Story in Transition

SCP-7b: Home . . . (pause), sounds like a good place. It will be nice to go back. They made this room feel like home with pictures from home. Are those pictures of your family?

P-7b: My family . . . and the story begins

### 4.2.3. Storytelling Option 3: Connect with Surrounding

SCP-7c: Back home . . . (pause), sounds like a good plan. The team here would like to know you better and help with the plan to go back home. I would like to know you too, and learn something from you before you go back home.

P-7c: (curious) What do you want to know?

(Multiple options)

### *4.3. Storytelling for Assessment*

During the duration of the two weeks, the SCP has become more familiar with Mr. C and learned some aspects of his story to help establish a rapport. The SCP has established a steady ground to move forward with the assessment. Assessment plays a key role in any discipline within the helping profession. Authors in spiritual care have formulated helpful assessment tools that guide the work of practitioners in health care (cf. Fitchett 1993; Pruyser 1976). In his book "Assessing spiritual needs", Fitchett (1993) reviews a model he and his team worked together to develop named 7X7 and three other models. Equipped with such tools, SCPs can employ storytelling to obtain the content for assessment. In the African oral tradition, I have observed such mediums as the use of a familiar song, story, or proverb to help make connections in the process of diagnosis. Questions such as what brings you here or what can you teach us may help SCPs determine how/where to proceed.

### 4.3.1. Spiritual Story

SCP-1 Hello Bob

P-1: Hello, who are you?

SCP-2: It is Florence, just here to say hello

P-2: Oh, yah. I saw you the other day. Do you work with these people?

SCP-3: Yes I do

P-3: What exactly do you do?

SCP-4: I am the lucky one, I get to sit down and listen to the stories of the day. Like your story, the story about your day.

P-4: My day is quiet, no story to tell. The Doctor tells me that I . . . That has been pretty much the story every day, nothing new. I expected that much

SCP-5: I hear you . . . (pause). I'm sorry that this has been part of your story and admire your resilience.

P-5: Resilience (deep thought and a smile). Sometimes I wonder why I'm still here. You see, I grew up . . .

The story continues

### 4.3.2. Option B: Assessing for Spiritual Coping

SCP-5: I hear you . . . (pause). How have you coped with that?

P-5: The story continues

### 4.3.3. Option C: Assessing for Spiritual Goals

SCP-5b: I hear you . . . (pause). That must be tough, huh
P-5b: Yes. It has been tough, but not always, . . . The story continues

### 4.4. *Storytelling for Intervention*

The goal of spiritual care, in this case, is to facilitate meaning-making through therapeutic storytelling. The SCP seeks to foster a conducive environment for a positive therapeutic relationship and provide Mr. C with space and time for his unfolding story. At this point, storytelling can be reciprocal. It is likely that the patient may start asking the SCPs for their own story, depending on the level of comfort and trust. It is helpful for the SCP to practice safe and effective use of self in their story sharing. The spontaneity of the African oral tradition, as observed in the practice of sharing testimonies in the early years of the church in Africa, can be of value in story sharing. Stories tend to spark memories and generate more stories from participants.

### 4.4.1. Incorporating Rituals and Objects

SCP-1a: Hello Bob, it is me. Florence
P-1a: Hello to you, come on in
SCP-2a: I come bearing gifts
P-2a: Gifts are good, what do you have
SCP-3a: I brought this songbook, from the Sanctuary
P-3a: Isn't that wonderful. It looks just like the one we used in the church, only better
SCP-4a: How come
P-4a: Back then, we . . . and the story continues followed by meaning making

### 4.4.2. Option B: Expanding Possibilities

SCP-1b: Hello Bob, it's Florence. Is this a good time for a visit?
P-1b: Oh yes, come on in
SCP2b: What are you up to this morning?
P-2b: A whole lot of nothing (smiling)
SCP-3b: Would like to try something?
P-3b: Like what?
SCP-4b: How about a short trip off unit, to the Sanctuary?
P-4b: That Sanctuary, . . . huh (smiling). Where is the Sanctuary
SCP-5b: It's just down the hallway from this unit, I can take you there
P-5b: You can (looks happy)?
SCP-6b: Oh, yes. I checked with the nurse and he said it was okay.
P-6b: Will someone tell my family? I think they are coming to visit today
SCP-7b: We will be back before they come. They will find you right here in your room
P-7b: Okay then, I am ready. Let us go see the Sanctuary
SCP-8b: (wheeling down the hallway) what's in your mind?
P-8b: It is good out here, just like it was. It was raining, when I . . . the story continues

### 4.4.3. Option C: Capitalizing on Presence

SCP-1c: Hello Bob, it's Florence (bringing nothing but open hands, open mind ready to listen and learn). May I come in? (in most cases, Bob will be sitting out in the lounge and the question will likely be "may I sit here with you?")
P-1c: Yes, you may. What do you want to do?
SCP-2c: I would like to sit here with you.
P-2c: It's okay, you can sit. If you see my son, tell him I'm sitting here. He can come right here not in my room.
SCP-3c: Yes, I will. I will look out for him
P-3c: He knows this place. His mother was here, but she did not come back home.
SCP-4c: I'm sorry (pause) . . . what happened

P-4c: I don't know. I think she is gone. (Silence) You know she planted those flowers, she loved flowers. She talked about flowers all the time.

SCP-5c: That sound beautiful, what was her name?

P-4c: "Elizabeth" but everyone called her "Betty"

SCP-5c: Betty, that is a good name

P-5c: (laughter) Yes it is a good name. I told her it was better than my name but she didn't believe me (more laughter). She was . . . (the story continues)

## 5. Theory Integration

In this case, the SCP sets the stage for storytelling as an approach to the provision of spiritual care and support. Storytelling tends to come naturally with older adults of any heritage and background. The two most important assets that older adults have is time and a reservoir of experience, information, and knowledge. Many of them are also aware of the increase in the pace of life over the years leading to an even busier contemporary life. They have lived through the economic growth and development that seemed to overshadow familial and social lives, leaving little to no time at all for social interactions or what may seem similar to casual small talk. Such casual small talks are likely to evolve into meaningful storytelling once individuals feel safe and comfortable enough to open up and share parts of their life stories. SCP needs to be intentional and spontaneous at setting the stage and adding value in spiritual conversations with patients/residents in health care and other related settings. Scott (2011) observes, "storytelling is a quintessentially social activity. It requires not only readers or listeners but other storytellers. Stories are at once the raw material and the cultural product of memory. Their telling creates a sense of immediacy (even when they are about very old events and actions) . . . and the universal feeling of taking part" (p. 205). Mitty (2010), too, highlights the connection between storytelling and old age when she writes "there is a connection between old age and spirituality and a quest for transcendence—to express one's self as part of the human condition". Mitty writes from a Nursing perspective and provides suggestions "on how to help older adults tell their stories, even if they are cognitively challenged by memory and language loss" (p. 58). The stories facilitate a meaningful connection and provide opportunities for older adults to add key details to their stories with each encounter. There are specific skills that may enhance the efforts of SCPs in their attempt to encourage storytelling in spiritual conversations and add value to the process of making meaning with older adults. I will outline three, recognizing that there are many other skills that practitioners can acquire through spiritual care education and training programs. This section will analyze three basic skills that practitioners can start with and continue to grow and other skills through their encounters with patients/residents and allow the experience to lead the process.

### 5.1. Sponteinity to Flow with the Rythym

Storytelling is likely to be more effective in less structured, more spontaneous, and informal environments. SCPs do enter patients'/residents' spaces to provide care. It is reasonable to expect that the therapeutic relationship is professional and formal. With older adults, some degree of informality, such as meeting outdoors by the garden, taking a stroll down the hallway or to the coffee shop, or joining the patient/resident as they watch a program or listen to music.

### 5.2. Attentive to the Flow of the Story

SCPs need to be attentive and intentional in spiritual conversation to allow an organic flow of the story, only helping with cues when and where necessary. There can be gaps in memory, and the patient's/resident's body language may communicate the need for help recalling certain general details about particular seasons, events, or places in time. Being attentive to the flow of the story may help SCP know when to offer such assistance.

*5.3. Affirming the Story*

SCPs play a key role in the storytelling/listening episode. A life/story shared is a life/story known and witnessed. Older adults can be encouraged with the satisfaction of knowing that someone, an unrelated individual, has known and witnessed their sacred story. In addition to rediscovering the meaning in their life/story or experience, the older adult is encouraged by the knowledge that their story is a source of meaning to another. Giving feedback, acknowledging the story, and expressing gratitude to the storyteller for their story can be affirming.

The context described in the case above is likely to occur in a health care or long-term care setting where the incoming individual is new to Staff. In gathering as much information as possible, the SCPs are likely to gain the most precise data for assessment and guarantee relevant and effective interventions. There are standardized steps that SCPs can follow to establish rapport and obtain information that informs ongoing assessment and interventions. Health systems have efficient processes in place for data collection to help make the work of health care professionals, including SCPs, efficient, seamless, and transferable in multiple contexts. These processes have been perfected over time with the necessary skills needed to succeed in the position and are made accessible through education and training. Theories of spiritual care include best practices in charting and documentation. The data that include demographic information, referral sources, assessment, and intervention is readily available for use by SCPs in clinical settings. Fitchett (1993) notes, "Assessment is both a statement of a perception and a process of information gathering and interpreting" (p. 17). He argues that assessment is both process and content. Since the publishing of that work, Fitchett has performed extensive work in spiritual assessment, some in collaboration with others (cf. Fitchett 2011; Nolan 2018). In a keynote speech titled: "Assessing spiritual needs in clinical setting" in 2009, Fitchett outlined assessment tools in spiritual care, including templates developed by Pulchaski in 1999. He developed the tool with the acronym FICS which stands for **F**aith-belief, **I**mportance, **C**ommunity, **A**ddress in care in taking a spiritual history. The American Family Physician Journal reprinted the document that discusses a tool with the acronym HOPE, representing; H-Sources of hope; O-organized religion; P-personal spirituality/practices; E-effect on medical care and end-of-life issues. These tools and others within the field are essential in spiritual care and should be considered complementary in the process of gathering information for assessment, especially in cases where the option of storytelling is significantly diminished. Such processes and tools equip SCPs with the resources needed to provide relevant and effective care. However, storytelling can go a long way in complementing standardized tools and processes to help locate the individual within the group. Encouraging patients/residents to narrate their story in their own words can empower the individual patient/resident, reassure them of their humanity and dignity and add value to their process of meaning-making. This practice can foster meaningful conversations with older adults.

## 6. Storytelling and Narrative Approach in Therapy: A Comparative Analysis

The narrative approach in family therapy has a distinct similarity with therapeutic storytelling. The theory falls under postmodern social construction family therapy; however, narratives in themselves are a universal phenomenon that has functioned across the ages as a healing force for different kinds of emotional trauma (Bergner 2007). Writing from the perspective of a therapist, Bergner asserts, "there is something about a good story, particularly one with personal relevance to the listener that gives it usually good staying power". He proceeds to state, "told well and used judiciously, stories stand out from the gestalt of the overall therapeutic conversation and tend not to be forgotten" (p. 5). There are similarities between the storytelling and narrative approach, such that a practitioner familiar with storytelling may be more inclined to work effectively with the narrative approach in therapy. Moen (2006) addressed the topic of narrative Research and used the words, stories, and narratives interchangeably in her work. She wrote:

One way of structuring these experiences is to organize them into meaningful units. One such meaningful unit could be a story, a narrative. For most people, storytelling is a natural way of recounting experience, a practical solution to a fundamental problem in life, creating reasonable order out of experience. Not only are we continually producing narratives to order and structure our life experiences, we are also constantly being bombarded with narratives from the social world we live in. We create narrative descriptions about our experiences for ourselves and others, and we also develop narratives to make sense of the behavior of others. (p. 60)

Reinstating her thesis of storytelling as an intercultural approach in pastoral care, Doehring (2015) claims, "caregivers can listen for underlying emotions that might point to a constellation of values, beliefs, and practices that make emotional/spiritual care" (p. 5). In their entirety, stories seem to have a quality of sufficiency, whether employed for simple entertainment or formal education. Bitter (2014) writes about the narrative approach stating that "narrative therapists approach self-language and self-talk much more holistically: People live "storied" lives, and their stories have contexts or backgrounds of interpreted experience, but they also intend and move toward anticipated outcomes in the future" (p. 311). Therapists seek to identify and shed a spotlight on family stories that control, contrasting them with stories that allow personal control. The case above described a situation with a patient who is living with specific health challenges that have caused a significant physical and cognitive decline. In such cases, I provide storytelling a chance to help gain relevant information in unfavorable conditions. In a case where the patient/resident is able and capable of engaging in a conversation and recalling details of their story with minimal to no assistance, the work of SCPs may feel much easier. The SCPs only need to be available, create an environment that can foster meaningful spiritual conversations, and allow themselves to receive and learn from the seasoned wisdom that is the human library. In the process, SCPs present patients/residents with the gift of their presence. As Van Katwyk (2003) writes, "it is important to encourage an approach that "critically assesses the impact of our personal and communal stories and, through therapeutic conversation, seeks to reinterpret, sometimes rewrite, closed and oppressive stories into alternative versions which open up to life and include our creative participation" (p. 369).

As Parry and Doan (1994) summarize in their discussion, therapeutic stories carry information about the familial traditions and meaning constructions of the particular people involved and also deconstruct and revise phases of therapy. As family tales told in the therapeutic environment, they help reveal the major sources of authorship that were influential in penning the scripts, which the family members have been invited to accept as "just the way things are". Robinson-Wood (2005) also observes, "at its core, storytelling has long served as a way to articulate what is, and how our present realities have come to be". When applied to counseling, storytelling facilitates the communication by the client about experiences in a language that is comfortable and familiar.

## 7. Conclusions

Storytelling is a distinct approach in the oral traditions that seems to be gaining interest both in research as narrative inquiry and in the practice of care. As much as the method is reclaiming space in therapeutic settings with diverse populations, it is most relevant when applied in spiritual conversations with older adults. This paper examined storytelling from the understanding of the African oral traditions to demonstrate its relevance in therapeutic conversations with older adults. Storytelling can be employed in spiritual conversations to complement other postmodern approaches in therapeutic settings to facilitate healing. Spiritual care practitioners can seek to be more spontaneous and intentional in their work with older adults and encourage them to share their stories in spiritual conversations that inform meaning-making. This method can find expression through the narrative approach in therapy. Using autoethnography, I analyzed a contrived verbatim spiritual conversation from a representative therapeutic encounter that employed a storytelling/listening

approach in spiritual care. The analysis helped demonstrate the relevance of including storytelling in spiritual conversations to enhance therapeutic encounters. The study also highlighted the apparent similarities between the traditional approach to storytelling and the postmodern narrative approach in therapy.

**Funding:** This research received no external funding.

**Institutional Review Board Statement:** Not applicable.

**Informed Consent Statement:** Not applicable. The study did not involve human or animal subjects.

**Data Availability Statement:** Not applicable.

**Conflicts of Interest:** The author declares no conflict of interest.

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
