# Peer review of "Recapturing the Oral Tradition of Storytelling in Spiritual Conversations with Older Adults: An Afro-Indigenous Approach"

_religions, doi:10.3390/rel13060563_

Round 1

Reviewer 1 Report

My two major concerns are:

  1. There are a number of typos- including spelling "samctuary," adn capitalization issues, that should have been addressed before submission.
  2. The quoted conversations, while interesting, don't, to this reader, make an compelling case as a comparison with an African oral tradition.  Some examples from African oral traditions placed side-by-side with the interviews from cases A,B,C and making the structural and spiritual similarities more overt would be very helpful.

Author Response

Thank you for taking time to review my manuscript and giving valuable feedback.  I have revised the manuscript and addressed the suggestions/recommendations.

I also want to apologize for the typo errors that I could have easily caught through the "Spell check" Function.  I have gone through the document and corrected all the errors

I reviewed the case and edited to analyze only one case.  I also added a case description

Counseling was assumed to be conveyed through the therapeutic conversation.  I have given two examples of African Oral Traditions and added the word "African" in relevant places to maintain consistency.

I added a section describing African Oral Tales to help demonstrate the relevance

I accidentally forgot to turn on the "Track changes" Function till much later into the revisions.  I apologize for the oversight.

Thanks you

Reviewer 2 Report

The manuscript is well-written, and shows interesting objectives,  scientifically relevant to palliative care, enhancing the value of African oral tradition as a therapeutic resource.

Errors in the spelling of several words throughout the text.

The conclusion is not deductible from the exposure preceding it. Examples of approaches using SCP are the highlight of the manuscript, but the results of the recourse to this technique were not mentioned.

Tip: add the results obtained with each of the approaches exposed as plausible to the patient, thus demonstrating the practical value of the use of PCS – as promised in topic 5: “The following is an analysis of three cases presenting typical scenarios prevalent among the adult older in geriatric clinical programs, retirement homes, and long term care facilities”

Topic 7 quotes: "Therapists seek to identify and shed a spotlight on family stories that control, contrasting them with stories that allow personal control. The case above described a situation with a patient who is living with specific health challenges that have caused a significant physical and cognitive decline. It was a deliberate choice to give storytelling a chance when conditions are unfavorable." However, the case was not described, only introduced without further details.

The same topic ends up arguing that "When applied to counseling, storytelling facilitates the communication by the client about the experience in a language that is comfortable and familiar", but the counseling offered to the patient was not previously reported.

To report some examples of tales of African oral tradition used during fieldwork, could clarify why knowledge of this tradition is as important and effective as the tales of other matrices of therapeutic narratives, also classified as storytelling, making the title fit the text.

The title cites Afro-indigenous oral tradition. But the text mentions only African tradition - and it is not clear which of the two terms would be appropriate, because not even the tales of oral tradition were explicit.

In the topic Conclusion, it is said "This paper examined storytelling from the understanding of the African Oral Traditions to demonstrate its relevance in therapeutic conversations with older adults" but no African tales reports were mentioned, and therefore no relevance was demonstrated.

Author Response

Thank you for taking time to read my manuscript and offering valuable feedback.  I have revised the manuscript following your feedback

I apologize for all the typo errors that I should have paid a closer attention to and caught with the use of a "spell check" function.  I have now checked through the 'spell check" and made necessary edits.

I have added examples of the oral traditions and edited the case example to capture only the one relevant case, removing cases A and B

In my eagerness to start revisions, I accidentally forgot to turn on the "Track Changes" Function till much later.  I apologize for this oversight.

Thanks you again for your time and feedback.

Reviewer 3 Report

The author clearly defined rationale for writing the paper and provided supporting evidence for its outcome.  The author succinctly describes how what is defined as spiritual is connected to storytelling which is part of oral traditions. In turn, storytelling is presented as therapeutic, especially for older adults. The author clearly delineates between ethnography and autoethnography which is a unique characteristic of the research. The paper is well written and shows how storytelling can be learned through education and training. This is useful information for healthcare professionals in therapeutic settings. In addition, older adults are portrayed in a positive light based on their life experiences and knowledge obtained. The paper makes a significant contribution to the literature on the oral tradition of storytelling.

Author Response

Thank you for taking time to read my manuscript and offering valuable feedback.  I have revised the manuscript following your feedback

I apologize for all the typo errors that I should have paid a closer attention to and caught with the use of a "spell check" function.  I have now checked through the 'spell check" and made necessary edits.

In my eagerness to start revisions, I accidentally forgot to turn on the "Track Changes" Function till much later.  I apologize for this oversight.

Thanks you again for your time and feedback.

Florence

Round 2

Reviewer 1 Report

Suggest a close proofread.

Reviewer 2 Report

Improvements done in the text explained why and how African oral tradition helped the researcher better achieve their goal. The results are good and relevant enough to be shared with the scientific community.